# Detection of Adulterated *Naodesheng* Tablet (*Naodesheng Pian*) via In-Depth Chemical Analysis and Subsequent Reconstruction of Its Pharmacopoeia Q-Markers

**DOI:** 10.3390/molecules29061392

**Published:** 2024-03-20

**Authors:** Chunhou Li, Xican Li, Jingyuan Zeng, Rongxin Cai, Shaoman Chen, Ban Chen, Xiaojun Zhao

**Affiliations:** 1School of Chinese Herbal Medicine, Guangzhou Higher Education Mega Center, Guangzhou University of Chinese Medicine, Waihuang East Road No. 232, Guangzhou 510006, China; 20211120209@stu.gzucm.edu.cn (C.L.); 20211110124@stu.gzucm.edu.cn (J.Z.); choi_roy@foxmail.com (R.C.); 20221110152@stu.gzucm.edu.cn (S.C.); zxj@gzucm.edu.cn (X.Z.); 2College of Pharmacy, Chengdu University of Traditional Chinese Medicine, Chengdu 611137, China; 3Key Laboratory of Fermentation Engineering, Cooperative Innovation Center of Industrial Fermentation, Ministry of Education & Hubei Province, Hubei University of Technology, Wuhan 430068, China; chenban@hbut.edu.cn

**Keywords:** counterfeiting recognition, *Naodesheng Pian*, quality control, UHPLC-Q-Exactive-Orbitrap MS/MS

## Abstract

*Naodesheng* Tablet (*Naodesheng Pian*), a traditional Chinese medicine formula for stroke treatment, is made up of five herbal medicines, i.e., *Sanqi*, *Gegen*, *Honghua*, *Shanzha*, and *Chuanxiong.* However, the current Pharmacopoeia quality-marker (Q-marker) system cannot detect possible adulteration. Our study tried to use a new strategy, i.e., standards-library-dependent ultra-high-performance liquid chromatography-quadrupole-Orbitrap mass spectrometry (UHPLC-Q-Orbitrap MS/MS) putative identification, to reconstruct the Q-marker system. Through the strategy, 30 isomers were successfully differentiated (such as 2′-hydroxygenistein, luteolin, and kaempferol; ginsenoside Rg2 and ginsenoside Rg3; ginsenoside Rf and ginsenoside Rg1). In particular, 11 compounds were unexpectedly found in *Naodesheng*, including 2′-hydroxygenistein, 7,4′-dihydroxyflavone, pectolinarigenin, 7-methoxy-4′-hydroxyisoflavone, scoparone, matrine, 3,3′,4′,5,6,7,8-heptamethoxyflavone, 5-hydroxyflavone, diosgenin, chloesteryl acetate, and (+)-4-cholesten-3-one. In total, 68 compounds were putatively identified and fully elucidated for their MS spectra. Subsequently, relevant compounds were further investigated using UV-vis scanning experiments, semi-quantitative analysis, and quantum chemical calculation. Finally, five adulterated *Naodesheng* Tablets were used for validation experiments. The experiment successfully detected five adulterated ones via a lower-version LC-MS analysis. On this basis, three new candidates (hydroxy safflor yellow A (HSYA), citric acid, and levistilide A), along with puerarin and notoginsenoside R1, are re-nominated as the Q-markers for LC-MS analysis. The LC-MS analysis of puerarin, notoginsenoside R1, HSYA, citric acid, and levistilide A can clearly detect adulteration regarding all five herbal medicines mentioned above. Therefore, the reconstructed Q-markers are described as a “perfect” quality control system to detect adulteration in *Naodesheng* and will offer a valuable recommendation for the Pharmacopoeia Commission.

## 1. Introduction

*Naodesheng* Tablet (*Naodesheng Pian*) is a traditional Chinese medicine (TCM) formula recorded in Chinese Pharmacopoeia (ChP). The Chinese “*Naodesheng*” means to promote the recovery from cerebral stroke, through activating blood circulation and removing blood stasis as well as clearing the channels. Therefore, it is widely consumed by numerous patients suffering from cerebral stroke [1]. Nowadays, there are 60 pharmaceutical factories manufacturing *Naodesheng* Tablet, according to the National Medical Products Administration of China [2]. 

The *Naodesheng* Tablet formula is made up of five herbal medicines, including *Gegen*, *Sanqi*, *Honghua*, *Shanzha*, and *Chuanxiong* (Table 1). ChP, however, has already defined its corresponding quality markers (Q-markers) when they were used as individual herbal medicines, that is, puerarin for *Gegen*, three saponins (ginsenoside Rg1 and Rb1, and notoginsenoside R1) for *Sanqi*, hydroxy safflor yellow A (HSYA) for *Honghua*, citric acid for *Shanzha*, and both levistilide A and ferulic acid for *Chuanxiong.* However, puerarin, HSYA, levistilide A, and citric acid have been excluded in the current *Naodesheng* Q-marker system for HPLC analysis (Table 1). 

This exclusion can cause two limitations. (1) Ferulic acid in *Naodesheng* cannot specifically characterize the presence of *Chuanxiong*, although ferulic acid is defined as the Q-marker of individual *Chuanxiong* by ChP. This is because ferulic acid is also enriched in other herbal medicines, e.g., *Honghua* [3] and *Shanzha* [4]. (2) ChP has tried to use a TLC tool to analyze puerarin and to characterize the presence of *Gegen* in *Naodesheng* [1]. However, this characterizing tool is highly tedious and the outcome is so unreliable because it relies on spot comparisons with the Rf value and blue color. The Rf value is well known to be variable and can be affected by external conditions. The blue color is actually a consequence of phenolic –OH interacting with FeCl_3_. Therefore, both Rf value and blue color do not have adequate specificity.

Two limitations further suggest that the current Pharmacopoeia Q-marker system can only specifically characterize *Sanqi*, because the system uses HPLC to analyze three *Sanqi*-derived saponins (ginsenoside Rg1, ginsenoside Rb1, and notoginsenoside R1). As a result, the other four herbal medicines lack specific Q-markers in *Naodesheng*, including *Gegen*, *Honghua*, *Shanzha*, and *Chuanxiong*. Therefore, the adulteration regarding four herbal medicines will not be detected by the current Pharmacopoeia Q-marker system. For example, if *Honghua* material is replaced by wood powder, adulterated Naodesheng Tablets will not be detected due to the lack of a *Honghua* Q-marker. A similar situation may also occur with Chuanxiong, Shanzha, and even *Gegen*. 

Now, it has become an inevitable tendency to use some new and high-accuracy technologies, e.g., ultra-high-performance liquid chromatography-quadrupole-Orbitrap mass spectrometry (UHPLC-Q-Orbitrap MS/MS), to reconstruct the Pharmacopoeia Q-marker system. The reconstruction requires a systematical investigation of bioactive compounds in *Naodesheng*. For this purpose, our study developed a reliable standards-library-dependent UHPLC-Q-Orbitrap MS/MS strategy.

The strategy depends on a set of authentic standards. After these standards were analyzed using UHPLC-Q-Orbitrap MS/MS, numerous and high-accuracy data were obtained and saved in the equipped software. Then, these data were used for matching with those from the *Naodesheng* sample, which also was subjected to a similar analysis under the same conditions. Through matching tests, the compounds from *Naodesheng* were putatively identified for their structures and even configurations.

Due to the high efficiency and high accuracy, the strategy is expected to offer reliable outcomes for compound identification. From these identified compounds, appropriate Q-marker candidates will be re-nominated based on experimental and theoretical approaches. Finally, the adulteration detection feasibility of these Q-marker candidates will be further verified by a lower-version LC-MS technology.

## 2. Results and Discussion

### 2.1. UHPLC-Q-Orbitrap MS Identification

Corresponding materials can be found in the following text (Section 3.3). LNT was processed into a sample solution. The LNT sample solution was subsequently assayed by means of the UHPLC-Q-Orbitrap MS/MS method. The total ion current (TIC) diagram is shown in Figure 1. Meanwhile, the main information on chromatographic peaks is detailed in Table 2. The information refers to retention time (R.T.) values, molecular ion peak, main MS/MS fragments, and documental evidence. Through comparison with corresponding authentic standards, 68 compounds were identified (Figure 2). The identification evidence is shown in Appendix A. The evidence indicates that the emerging UHPLC-Q-Orbitrap MS analysis was much more effective than previous HPLC-UV analyses [5,6,7,8] because the emerging analysis could simultaneously determine hundreds of compounds.

Its high efficiency was further supported by Wu’s work which simultaneously identified 189 compounds from *Bufei Yishen* Formula. However, Wu’s work failed to offer a full MS spectrum elucidation of all compounds and also to distinguish isomers [9]. Therefore, Wu’s work could only be considered as a tentative identification and our work as a putative identification.

Our putative identification, however, has been documented to possess evident advantages in MS spectrum elucidation and isomer distinction [10]. These advantages could also be found in the present study. As seen in Appendix A, all 68 compounds have been elucidated for their MS spectra based on fragmenting principles. The elucidation revealed that there were only 10^−7^ RSD values between the calculated and experimental *m*/*z* values. For example, a *m*/*z* 391 peak in Appendix A was calculated as 2.6 × 10^−7^ RSD (391.2848 vs. 391.2850). Such a low RSD value has suggested our identification to be highly reliable. Moreover, the error values (δ) between experimental *m*/*z* values and theoretical *m*/*z* values of the molecular ions of all identified compounds were also calculated and are listed in Table 2. 

In contrast, the previous tentative identification could not offer MS spectrum elucidation and thus had to cite outdated documental data to match their experimental ones [11,12]. The identification of calycosin was a typical instance. Its positive model peaks (*m*/*z* 285, 270, and 134) were used to match the negative model peak values (*m*/*z* 283, 268, and 239). There is obviously no comparability between the two groups of data in *m*/*z* values, determination models, and apparatus conditions; correspondingly, the previous study could not offer MS spectrum elucidation and only listed the MS spectrum *m*/*z* values [11,12].

Our second advantage was isomer distinction; this was based on our new method [13]. Following the new method and depending on a standards library, our study successfully differentiated 30 isomers from each other (Figure 2A), under the same UHPLC-Q-Orbitrap MS analysis condition. These differentiated isomers are 2′-hydroxygenistein, luteolin, and kaempferol; ferulic acid and isoferulic acid; daidzein and 7,4′-dihydroxyflavone; genistein and apigenin; 7-methoxy-4′-hydroxyisoflavone and formononetin; calycosin and prunetin; pratensein and diosmetin; chlorogenic acid and cryptochlorogenic acid; 3′-hydroxy puerarin and genistin; hyperoside and isoquercitrin; daidzin and puerarin; ginsenoside Rg2 and ginsenoside Rg3; and ginsenoside Rf and ginsenoside Rg1.

The distinction of three isomers 2′-hydroxygenistein, luteolin, and kaempferol was a typical instance. As illustrated in Appendix A, the three possessed the same [M − H] peak (*m*/*z* 285); however, their MS/MS peaks were different from each other. Another typical instance was the distinction of ginsenoside Rg1 and its isomer ginsenoside Rf. As illustrated in Appendix A, two isomers displayed identical [M − H] peals (*m*/*z* 799) and similar diagnostic MS/MS peaks (*m*/*z* 637, 475, and 391). However, their MS/MS profiles and R.T. values were different from each other. Accordingly, two isomers were clearly differentiated (Figure 2A and Appendix A). Similar to the pair of ginsenoside Rg1 and ginsenoside Rf, the pair of ginsenoside Rg2 and ginsenoside Rg3 was also differentiated depending on the MS/MS peak fragments. As seen in Appendix A, ginsenoside Rg2 showed diagnostic fragments at *m*/*z* 637, 619, 475, and 391, while its isomer ginsenoside Rg3 displayed diagnostic fragments at *m*/*z* 621 and 375. According to the different diagnostic fragments, ginsenoside Rg2 and ginsenoside Rg3 were also differentiated from each other. By comparison, previous studies have not distinguished these isomers and had to use ambiguous phrases, such as “ginsenoside Rg2 or isomer”, “isomer”, or “dimer”, to describe the identification outcomes [11,12,14,15,16,17,18,19,20,21,22,23,24,25].

**Table 2 molecules-29-01392-t002:** The main information of 68 putatively identified bioactive compounds (**1**~**68**) from *Naodesheng* Tablet.

ID	R.T. min	Name	Molecular Ion	Experimental *m*/*z* Value	Theoretical *m*/*z* Value	Error δ (ppm)	Diagnostic Fragments *m*/*z*	Plant Resource
**1**	0.53	D-gluconic acid	C_6_H_11_O_7_^-^	195.0506	195.0510	2.05	177.0396, 159.0295, 129.0182	*Sanqi* [26]
**2**	0.58	citric acid	C_6_H_7_O_7_^-^	191.0192	191.0192	0.00	173.0078, 129.0184, 111.0077	*Shanzha* [1]
**3**	1.17	*L*-phenylalanine	C_9_H_10_NO_2_^-^	164.0709	164.0712	1.83	148.0777, 147.0446, 103.0540	*Sanqi* [26]
**4**	1.56	protocatechuic acid	C_7_H_5_O_4_^-^	153.0182	153.0188	3.92	110.0316, 109.0290, 108.0211	*Chuanxiong* *Shanzha* [27]
**5**	1.73	*L*-tryptophan	C_11_H_11_N_2_O_2_^-^	203.0821	203.0821	0.00	186.0546, 159.0918, 142.0651, 116.0494	*Chuanxiong* [28]
**6**	3.68	chlorogenic acid	C_16_H_17_O_9_^-^	353.0883	353.0873	2.83	191.0556, 173.0450, 161.0237, 127.0395	*Shanzha* [27]
**7**	3.95	HSYA	C_27_H_31_O_16_^-^	611.1616	611.1612	0.65	491.1191, 403.1029, 325.0712, 283.0597, 119.0492	*Honghua* [29,30]
**8**	4.33	vanillic acid	C_8_H_7_O_4_^-^	167.0349	167.0344	2.99	152.0104, 123.0439, 108.0204	*Chuanxiong* [28]
**9**	4.40	caffeic acid	C_9_H_7_O_4_^-^	179.0343	179.0344	0.56	136.0473, 135.0446, 117.0334, 107.0496	*Chuanxiong* [28]
**10**	4.50	cryptochlorogenic acid	C_16_H_17_O_9_^-^	353.0867	353.0873	1.70	191.0556, 179.0348, 173.0445, 135.0446	*Chuanxiong* [28]
**11**	5.89	3′-hydroxy puerarin	C_21_H_19_O_10_^-^	431.0985	431.0978	1.62	311.0556, 283.0606, 255.0657, 227.0708	*Gegen* [31]
**12**	7.94	puerarin	C_21_H_19_O_9_^-^	415.1038	415.1029	2.17	295.0611, 267.0657, 253.0512, 132.0211	*Gegen* [31]
**13**	8.23	3′-methoxy puerarin	C_22_H_19_O_10_^-^	445.1138	445.1135	0.67	325.0713, 282.0534, 253.0509, 225.0551, 148.0155	*Gegen* [31]
**14**	8.39	mirificin	C_26_H_27_O_13_^-^	547.1447	547.1452	0.91	325.0712, 295.0606, 267.0657, 132.0205	*Gegen* [31]
**15**	8.47	daidzin	C_21_H_19_O_9_^-^	415.1029	415.1029	0.00	252.0421, 223.0395, 195.0446, 167.0493	*Gegen* [31]
**16**	8.57	ferulic acid	C_10_H_9_O_4_^-^	193.0506	193.0501	2.59	178.0261, 149.0579, 137.0239, 134.0362	*Chuanxiong* [28], *Honghua* [3] *Shanzha* [4]
**17**	8.66	isoferulic acid	C_10_H_9_O_4_^-^	193.0498	193.0501	1.55	178.0261, 149.0579, 137.0239, 134.0362	*Honghua* [3]
**18**	8.72	glycitin	C_22_H_21_O_10_^-^	445.1143	445.1136	1.57	325.0727, 267.0300, 239.0345, 211.0395	*Sanqi* [26]
**19**	9.16	genistin	C_21_H_19_O_10_^-^	431.0978	431.0978	0.00	268.0372, 239.0344, 211.0395, 195.0446	*Gegen* [31]
**20**	9.23	4-methyl-2,6-dimethoxyphenol	C_9_H_11_O_3_^-^	169.0861	169.0865	2.37	137.0592111.0446, 109.0653, 107.0497	*Honghua* [3] *Chuanxiong* [32]
**21**	9.42	hyperoside	C_21_H_19_O_12_^-^	463.0873	463.0877	0.86	300.0268, 271.0244, 255.0293, 243.0293	*Shanzha* [27], *Chuanxiong* [28]
**22**	9.50	rutin	C_27_H_29_O_16_^-^	609.1461	609.1456	0.82	300.0269, 271.0244, 255.0292, 243.0291	*Honghua* [3] *Shanzha* [27]
**23**	9.55	isoquercitrin	C_21_H_19_O_12_^-^	463.0877	463.0877	0.00	300.0269, 271.0244, 255.0293, 243.0293	*Shanzha* [27]
**24**	9.66	*S*-naringin	C_27_H_31_O_14_^-^	579.1703	579.1314	6.17	271.0612, 151.0025, 119.0497, 107.0126	*Gegen* [33]
**25**	9.77	cosmosiin	C_21_H_19_O_10_^-^	431.0981	431.0978	0.70	268.0377, 211.0395, 151.0031, 130.0410, 117.0340	*Chuanxiong* [32]
**26**	9.97	astragalin	C_21_H_19_O_11_^-^	447.0924	447.0927	0.67	327.0495, 284.0321, 255.0293, 227.0341	*Honghua* [3], *Gegen* [33], *Chuanxiong* [28]
**27**	10.23	2′-hydroxygenistein	C_15_H_9_O_6_^-^	285.0339	285.0359	7.02	217.0502, 199.0390, 149.0233, 133.0283	*Gegen* [34]
**28**	10.47	daidzein	C_15_H_9_O_4_^-^	253.0505	253.0501	1.58	223.0395, 208.0528, 195.0446, 180.0575	*Gegen* [31]
**29**	10.59	calycosin	C_16_H_11_O_5_^-^	283.0613	283.0606	2.47	268.0372, 239.0347, 211.0395, 195.0446	*Honghua* [35]
**30**	10.64	quercetin	C_15_H_9_O_7_^-^	301.0353	301.0348	1.66	245.0445, 151.0025, 139.0391, 121.0283	*Honghua* [3], *Shanzha* [27], *Gegen* [34]
**31**	10.67	7,4′-dihydroxyflavone	C_15_H_9_O_4_^-^	253.0504	253.0501	1.19	223.0395, 195.0446, 180.0571, 117.0340	*Gegen* [33]
**32**	10.68	syringic acid	C_9_H_9_O_5_^-^	197.045	197.0450	0.00	182.0210, 166.9975, 153.0548, 138.0311, 123.0076	*Honghua* [3] *Chuanxiong* [32]
**33**	10.70	pectolinarigenin	C_17_H_13_O_6_^-^	313.0718	313.0712	1.92	298.0482, 283.0243, 255.0293, 227.0334	*Gegen* [36]
**34**	10.86	luteolin	C_15_H_9_O_6_^-^	285.0404	285.0399	1.75	257.0434, 241.0492, 199.0391, 133.0283	*Shanzha* [27]
**35**	11.00	genistein	C_15_H_9_O_5_^-^	269.0457	269.0450	2.60	224.0471, 213.0553, 201.0552, 133.0285	*Gegen* [31]
**36**	11.01	notoginsenoside R1	C_47_H_79_O_18_^-^	931.5266	931.5266	0.00	799.4864, 637.4324, 475.3787, 391.2855	*Sanqi* [26,37,38]
**37**	11.09	pratensein	C_16_H_11_O_6_^-^	299.0506	299.0556	16.72	284.0327, 255.0293, 227.0344, 211.0395	*Gegen* [33]
**38**	11.20	diosmetin	C_16_H_11_O_6_^-^	299.0561	299.0556	1.67	284.0322, 256.0372, 227.0341, 183.0441	*Gegen* [33]
**39**	11.30	ginsenoside Rg1	C_42_H_71_O_14_^-^	799.4788	799.4844	7.00	637.4324, 475.3783, 391.2832, 179.0551	*Sanqi* [26,38]
**40**	11.39	apigenin	C_15_H_9_O_5_^-^	269.045	269.0450	0.00	241.0501, 225.0552, 213.0558, 117.0334	*Honghua* [3] *Shanzha* [27]
**41**	11.53	isoliquiritigenin	C_15_H_11_O_4_^-^	255.0657	255.0657	0.00	213.0552, 135.0076, 119.0497	*Gegen* [39]
**42**	11.78	7-methoxy-4′-hydroxyisoflavone	C_16_H_11_O_4_^-^	267.0664	267.0657	2.62	252.0423, 223.0395, 195.0446, 132.0206	*Gegen* [33]
**43**	11.82	kaempferol	C_15_H_9_O_6_^-^	285.0403	285.0399	1.40	255.0293, 229.0501, 211, 0392, 117.0340	*Honghua* [1,3], *Sanqi* [40], *Shanzha* [27]
**44**	11.87	formononetin	C_16_H_11_O_4_^-^	267.0660	267.0657	1.12	252.0426, 223.0395, 195.0446, 132.0208	*Gegeng* [31]
**45**	12.26	ginsenoside Rf	C_42_H_71_O_14_^-^	799.4831	799.4844	1.63	637.4299, 475.3781, 391.2848, 161.0450	*Sanqi* [26]
**46**	12.35	*20R*-notoginsenoside R2	C_41_H_69_O_13_^-^	769.4735	769.4740	0.65	637.4312, 475.3795, 391.2855, 161.0445	*Sanqi* [26]
**47**	12.36	prunetin	C_16_H_11_O_5_^-^	283.0612	283.0606	2.12	268.0372, 239.0334, 211.0395, 195.0446	*Gegen* [33,34]
**48**	12.52	ginsenoside Rg2	C_42_H_71_O_13_^-^	783.4887	783.4895	1.02	637.4316, 619.4217, 475.3784, 391.2850	*Sanqi* [26]
**49**	12.56	*20S*-ginsenoside Rh1	C_36_H_61_O_9_^-^	637.4323	637.4316	1.10	475.3780, 391.2863, 161.0448, 113.0234	*Sanqi* [26]
**50**	12.95	ginsenoside Rb1	C_54_H_91_O_23_^-^	1107.5951	1107.5951	0.00	945.5407, 783.4895, 621.4379, 459.3838	*Sanqi* [26,38]
**51**	12.97	8-prenyldaidzein	C_20_H_17_O_4_^-^	321.1131	321.1127	1.25	266.0579, 237.0552, 209.0603, 143.0493	*Gegen* [39]
**52**	13.74 *	ginsenoside Rd	C_48_H_83_O_18_^+^	945.5411	945.5423	1.27	783.4895, 621.4366, 161.0450	*Chuanxiong* [28]
**53**	14.31 *	ginsenoside Rg3	C_42_H_73_O_13_^+^	783.4877	783.4895	2.30	621.4366, 375.2899, 161.0450, 113.0239	*Sanqi* [26]
**54**	16.20 *	ethyl stearate	C_20_H_41_O_2_^+^	311.2955	311.2950	1.61	183.0111, 133.0654, 119.0491	*Sanqi* [26] *Honghua* [35]
**55**	0.88 *	matrine	C_15_H_25_N_2_O^+^	249.1952	249.1967	6.02	247.1801, 218.1544, 190.1227, 176.1052	*Gegen* [33]
**56**	1.17 *	5-hydroxymethylfurfural	C_6_H_7_O_3_^+^	127.0391	127.0395	3.15	109.0288, 97.0284, 81.0339, 69.0341	*Sanqi* [26]
**57**	5.47 *	caffeine	C_8_H_11_N_4_O_2_^+^	195.0874	195.0882	4.10	138.0667, 123.0428, 110.0718, 108.0562,	*Sanqi* [26]
**58**	9.29*	1,5-dicaffeoylquinic acid	C_25_H_25_O_12_^+^	515.1158	515.1190	6.21	353.0871, 335.0760, 191.0551, 135.0446	*Shanzha* [27]
**59**	9.37 *	scoparone	C_11_H_11_O_4_^+^	207.0646	207.0652	2.90	191.0334, 163.0388, 151.0759, 146.0360	*Chuanxiong* [32]
**60**	11.98 *	*S*-senkyunolide A	C_12_H_17_O_2_^+^	193.1213	193.1229	8.28	175.1123, 147.1167, 137.0603, 105.0704	*Chuanxiong* [28]
**61**	12.59 *	*Z*-ligustilide	C_12_H_15_O_2_^+^	191.1063	191.1072	4.71	173.0603, 145.1017, 129.0704, 115.0548	*Chuanxiong* [28]
**62**	12.68 *	3,3′,4′,5,6,7,8-heptamethoxyflavone	C_22_H_25_O_9_^+^	433.1481	433.1499	4.16	418.1254, 403.1014, 165.0552, 107.0496	*Gegen* [33]
**63**	12.88 *	tangeretin	C_20_H_21_O_7_^+^	373.1271	373.1287	4.29	358.1053, 343.0818, 297.0754, 271.0603,	*Gegen* [33]
**64**	13.27 *	5-hydroxyflavone	C_15_H_11_O_3_^+^	239.07	239.0708	3.35	221.0603, 137.0232, 129.0340, 103.0548	*Gegen* [33]
**65**	13.66 *	levistilide A	C_24_H_29_O_4_^+^	381.2084	381.2066	4.72	191.1067, 149.0593, 135.0442, 117.0702	*Chuanxiong* [28]
**66**	14.47 *	diosgenin	C_27_H_43_O_3_^+^	415.3198	415.3212	3.37	271.2050, 253.1940, 171.1174, 157, 1011	*Chuanxiong* [41]
**67**	16.22 *	chloesteryl acetate	C_29_H_49_O_2_^+^	429.3723	429.3727	0.93	401.3405, 205.1222, 165.0909, 105.0701	***Gegen*** [34]
**68**	16.67 *	(+)-4-cholesten-3-one	C_27_H_45_O^+^	385.3459	385.3470	2.85	367.3365, 173.1321, 123.0807, 109.0653	***Gegen*** [34]

Note: The peaks with *m*/*z* < 50 were also found by the Xcalibur 4.1 Software package, although the scanning mode range was set at *m*/*z* 100–1200 in the mass spectra. All identification processes, including MS elucidation, are detailed in Appendix A. R.T. values with “*” were detected in positive ion mode, while R.T. values without “*” were detected in negative ion mode. The error values (δ) were calculated using the formula δ = | experimental *m*/*z* value − theoretical *m*/*z* value | ÷ theoretical *m*/*z* value ÷ 10^−6^.

The above advantages have indicated our standards-library-dependent UHPLC-Q-Orbitrap MS putative identification to be of not only high efficiency but also high accuracy. By means of this putative identification, 11 unexpected compounds were found from *Naodesheng* Tablet for the first time, including 2′-hydroxygenistein, 7,4′-dihydroxyflavone, pectolinarigenin, 7-methoxy-4′-hydroxyisoflavone, scoparone, matrine, 3,3′,4′,5,6,7,8-heptamethoxyflavone, 5-hydroxyflavone, diosgenin, chloesteryl acetate, and (+)-4-cholesten-3-one. In fact, none of the documents suggested that these compounds were from *Naodesheng* or its relevant plants. This obviously supplied new chemical information regarding *Naodesheng.*

All these expected and unexpected compounds have actually created a premise to reconstruct the Pharmacopoeia adulteration detection Q-marker system. Consulting with the “five basic principles” of Academician Chang-xiao Liu [25,42] and considering that citric acid (**2**), HSYA (**7**), puerarin (**12**), notoginsenoside R1 (**36**), and levistilide A (**65**) have already acted as Pharmacopoeia Q-markers for individual herbal medicines (Table 1), our study thus re-nominated these compounds (**2**, **7**, **12**, **36**, and **65**) as new Q-markers (Table 3). The reason why the current Pharmacopoeia Q-markers system excluded citric acid (**2**), HSYA (**7**), and puerarin (**12**) may be attributed to the defects of HLPC-UV.

### 2.2. UV-Vis Spectrum Scanning and Computational Chemistry Results

To offer further evidence, five Q-marker candidates, citric acid (**2**), HSYA (**7**), puerarin (**12**), notoginsenoside R1 (**36**), and levistilide A (**65**), along with two old Q-markers (ginsenoside Rg1 **39** and ginsenoside Rb1 **50**), were scanned for UV-vis spectra. As seen in Figure 3, the five formed a complicated mixture and usually shared the same maximum absorption wavelengths. Even at a range of absorption wavelengths, such as 203, 250, and 325 nm, the detected compounds were limited to several main high-abundance compounds, including puerarin (**12**), notoginsenoside R1 (**36**), ginsenoside Rg1 (**39**), ginsenoside Rb1 (**50**), and HSYA (**7**) [6,7,8,43,44,45]. This greatly limited the selectivity when monitored by a UV-vis detector. On the other hand, their molecular polarities (i.e., dipole moment values) were close to each other (e.g., HSYA **7**, notoginsenoside R1 **36**, and levistilide A **65**, Table 4). As a result, they could not be effectively separated by a polarity-based adsorption chromatographic column (e.g., C_18_). All these findings from UV-vis spectrum scanning and computational chemistry suggest that conventional HPLC-UV was not applicable for the simultaneous analysis of five Q-markers.

### 2.3. Adulteration Detection Validation Experiment Based on Five Adulterated Naodesheng Tablets and Low-Version LC-MS

To verify whether the LC-MS technology was applicable for the simultaneous analysis of five Q-marker candidates, this study introduced low-version LC-MS (i.e., UHPLC-ESI-Q-TOF-MS) to analyze CNT 1~CNT 5. As seen in Figure 4A, the UHPLC-ESI-Q-TOF-MS analysis of normal *Naodesheng* Tablet clearly displayed a puerarin (**12**) peak at R.T. 1.375 min; however, the adulterated *Naodesheng* Tablet (CNT 1) had no peak at the corresponding site. The comparison suggested the absence of puerarin (**12**) and further indicated the adulteration of *Gegen* in *Naodesheng* Tablet. Similarly, the comparison between the two diagrams in Figure 4C evidently illustrates that HSYA (**7**) was absent in adulterated *Naodeshen* (CNT 3) and thus, *Honghua* was adulterated in *Naodeshen*. Similar successful instances can also be observed in Figure 4B,D,E. Apparently, these successes could be attributed to the high selectivity of the molecular formula extraction technology in LC-MS [46]^.^

Meanwhile, these successful experiments also showed that (1) the LC-MS technology was applicable for the analysis of these Q-markers. (2) More importantly, the adulteration regarding all five herbal medicines (*Sanqi*, *Gegen*, *Honghua*, *Shanzha*, and *Chuanxiong*) in *Naodeshen* could be effectively detected. Therefore, the reconstructed adulteration detection Q-marker system was described as a “perfect” one; it would provide valuable consideration for the ChP commission.

Finally, it should be noted that (1) ferulic acid cannot specifically characterize any herbal medicines because it is also distributed in *Chuanxiong* [28], *Honghua* [3], and *Shanzha* [4]; regardless, it has been used as a Q-marker of individual *Chuanxiong* (Table 1). (2) The reconstruction of the Q-marker system was based on the analysis of one batch of *Naodesheng* Tablets in our study. However, these Q-markers were also found in other batches by the previous ones [44,47,48] or Pharmacopoeia itself [1]. (3) Although *Naodesheng* Tablet was reported to be related to the repair of β-amyloid-induced dysfunction [49], the present study does not discuss these bio-pharmacological issues. In fact, the role of β-amyloid is still controversial nowadays [50].

## 3. Materials and Methods

### 3.1. Medicine Materials

*Naodesheng* Tablet (Lot. 210803) was manufactured by Harbin Huayu Pharmaceutical Co., Ltd. (Wuhan, China). *Gegen* (Lot. 201101) and *Shanzha* (Lot. 220702) were purchased from Anhui Huifeng Traditional Chinese Medicine Co., Ltd. (Bozhou, China); Chuanxiong (Lot. 221100381) was purchased from Kangmei Traditional Chinese Medicine Slices Co., Ltd. (Shantou, China); *Honghua* (Lot. 230303) was purchased from Putianhe Traditional Chinese Medicine Co., Ltd. (Anguo, China); *Sanqi* (Lot. 230601) was purchased from Hongya County Wawushan Pharmaceutical Co., Ltd. (Hongya, China).

Five adulterated *Naodesheng* Tablets were prepared by our team through the replacement method. *Gegen* was replaced by wood powder to prepare the first adulterated *Naodesheng* Tablet, i.e., CNT 1. Similarly, *Sanqi* was replaced by wood powder to obtain CNT 2. In addition, *Honghua*, *Shanzha*, and *Chuanxiong* were replaced by wood powder to produce CNT 3, CNT 4, and CNT 5, respectively.

### 3.2. Authentic Standards and Chemicals

Chlorogenic acid (C_16_H_18_O_9_, M.W. 354.31, Cas. 327-97-9, 98%), caffeic acid (C_8_H_8_O_4_, M.W. 180.16, Cas. 331-39-5, 98%), cryptochlorogenic acid (C_16_H_18_O_9_, M.W. 354.311, Cas. 905-99-7, 98%), mirificin (C_26_H_28_O_13_, M.W. 548.49, Cas. 103654-50-8, 98%), daidzin (C_21_H_20_O_9_, M.W. 416.38, Cas. 552-66-9, 98%), isoferulic acid (C_10_H_10_O_4_, M.W. 194.18, Cas. 537-73-5, 98%), genistin (C_21_H_20_O_10_, M.W. 432.37, Cas. 529-59-9, 98%), 4-methyl-2,6-dimethoxyphenol (C_9_H_12_O_3_, M.W. 168.19, Cas. 6638-05-7, 98%), hyperoside (C_21_H_20_O_12_, M.W. 464.37, Cas. 482-36-0, 98%), rutin (C_27_H_30_O_16_, M.W. 610.52, Cas. 153-18-4, 98%), isoquercitrin (C_21_H_20_O_12_, M.W. 464.38, Cas. 482-35-9, 98%), S-naringin (C_27_H_32_O_14_, M.W. 580.53, Cas. 10236-47-2, 98%), astragalin (C_21_H_20_O_11_, M.W. 448.38, Cas. 480-10-4, 98%), calycosin (C_16_H_12_O_5_, M.W. 284.27, Cas. 20575-57-9, 98%), quercetin (C_15_H_10_O_7_, M.W. 302.23, Cas. 117-39-5, 98%), 7,4′-dihydroxyflavone (C_15_H_10_O_4_, M.W. 254.24, Cas. 2196-14-7, 98%), syringic acid (C_9_H_10_O_5_, M.W. 198.17, Cas. 530-57-4, 98%), pectolinarigenin (C_17_H_14_O_6_, M.W. 314.29, Cas. 520-12-7, 98%), diosmetin (C_16_H_12_O_6_, M.W. 300.26, Cas. 520-34-3, 98%), apigenin (C_15_H_10_O_5_, M.W. 270.24, Cas. 520-36-5, 98%), isoliquiritigenin (C_15_H_12_O_4_, M.W. 256.25, Cas. 961-29-5, 98%), 7-methoxy-4′-hydroxyisoflavone (C_16_H_12_O_4_, M.W. 268.27, Cas. 486-63-5, 98%), 8-prenyldaidzein (C_20_H_18_O_4_, M.W. 322.35, Cas. 135384-00-8, 98%), 1,5-dicaffeoylquinic acid (C_25_H_24_O_12_, M.W. 516.45, Cas. 30964-13-7, 98%), tangeretin (C_20_H_20_O_7_, M.W. 372.37, Cas. 481-53-8, 98%), and diosgenin (C_27_H_42_O_3_, M.W. 416.40, Cas. 512-04-9, 98%) were purchased from Chengdu Alfa Biotechnology Co., Ltd. (Chengdu, China). Citric acid (C_6_H_8_O_7_, M.W. 192.12, Cas. 77-92-9, 98%), hydroxy safflor yellow A (C_27_H_32_O_16_, M.W. 612.53, Cas. 78281-02-4, 98%), 3′-methoxy puerarin (C_22_H_22_O_10_, M.W. 446.40, Cas. 117047-07-1, 98%), glycitin (C_22_H_22_O_10_, M.W. 446.40, Cas. 40246-10-4, 98%), cosmosiin (C_21_H_20_O_10_, M.W. 432.38, Cas. 578-74-5, 98%), 20R-notoginsenoside R2 (C_41_H_70_O_13_, M.W. 770.99, Cas. 948046-15-9, 98%), 20S-ginsenoside Rh1 (C_36_H_62_O_9_, M.W. 638.88, Cas. 63223-86-9, 98%), matrine (C_15_H_24_N_2_O, M.W. 248.37, Cas. 519-02-8, 98%), 5-hydroxymethylfurfural (C_6_H_6_O_3_, M.W. 126.11, Cas. 67-47-0, 98%), scoparone (C_11_H_10_O_4_, M.W. 206.19, Cas. 120-08-1, 98%), *S*-senkyunolide A (C_12_H_16_O_2_, M.W. 192.25, Cas. 63038-10-8, 98%), Z-ligustilide (C_12_H_14_O_2_, M.W. 190.24, Cas. 81944-09-4, 98%), and levistilide A (C_24_H_28_O_4_, M.W. 380.484, Cas. 88182-33-6, 98%) were purchased from Baoji Herbest Bio-Tech Co., Ltd. (Baoji, China). Protocatechuic acid (C_7_H_6_O_4_, M.W. 154.12, Cas. 99-50-3, 98%), puerarin (C_21_H_20_O_9_, M.W. 416.38, Cas. 3681-99-0, 98%), ginsenoside Rf (C_42_H_72_O_14_, M.W. 801.00, Cas. 52286-58-5, 98%), ginsenoside Rg2 (C_42_H_72_O_13_, M.W. 785.01, Cas. 52286-74-5, 98%), ginsenoside Rb1 (C_54_H_92_O_23_, M.W. 1109.29, Cas. 41753-43-9, 98%), ginsenoside Rd (C_48_H_82_O_18_, M.W. 963.17, Cas. 52705-93-8, 98%), ginsenoside Rg3 (C_42_H_72_O_13_, M.W. 785.01, Cas. 14197-60-5, 98%), and 3,3′,4′,5,6,7,8-heptamethoxyflavone (C_22_H_42_O_9_, M.W. 432.42, Cas. 1178-24-1, 98%) were purchased from Sichuan Weikeqi Biological Technology Co., Ltd. (Chengdu, China). 2′-Hydroxygenistein (C_15_H_10_O_6_, M.W. 286.23, Cas. 1156-78-1, 98%), luteolin (C_15_H_10_O_6_, M.W. 286.24, Cas. 491-70-3, 98%), notoginsenoside R1 (C_47_H_80_O_18_, M.W. 933.14, Cas. 80418-24-2, 98%), ginsenoside Rg1 (C_42_H_72_O_14_, M.W. 801.02, Cas. 22427-39-0, 98%), formononetin (C_16_H_12_O_4_, M.W. 268.26, Cas. 485-72-3, 98%), and prunetin (C_16_H_12_O_5_, M.W. 284.26, Cas. 552-59-0, 98%) were purchased from BioBioPha Co., Ltd. (Kunming, China). D-Gluconic acid (C_6_H_12_O_7_, M.W. 196.16, Cas. 526-95-4, 98%), vanillic acid (C_8_H_8_O_4_, M.W. 168.15, Cas. 121-34-6, 98%), ethyl stearate (C_20_H_40_O_2_, M.W. 312.53, Cas. 111-61-5, 98%), and chloesteryl acetate (C_29_H_48_O_2_, M.W. 428.69, Cas. 604-35-3, 98%) were purchased from Sigma-Aldrich Co., Ltd. (Shanghai, China). 5-Hydroxyflavone (C_15_H_10_O_3_, M.W. 238.24, Cas. 491-78-1, 98%), genistein (C_15_H_10_O_5_, M.W. 270.24, Cas. 446-72-0, 98%), and (+)-4-cholesten-3-one (C_27_H_44_O, M.W. 394.55, Cas. 601-57-0, 98%) were purchased from TCI Chemical Co., Ltd. (Shanghai, China). L-Phenylalanine (C_9_H_11_NO_2_, M.W. 178.18, Cas. 63-91-2, 98%) and L-tryptophan (C_11_H_12_N_2_O_2_, M.W. 204.23, Cas. 73-22-3, 98%) were obtained from J&K Scientific Co., Ltd. (Beijing, China). Daidzein (C_15_H_10_O_4_, M.W. 254.24, Cas. 486-66-8, 98%) and caffeine (C_8_H_10_N_4_O_2_, M.W. 194.19, Cas. 58-08-2, 98%) were obtained from Chengdu Biopurify Phytochemicals Co., Ltd. (Chengdu, China). Kaempferol (C_15_H_10_O_6_, M.W. 286.24, Cas. 520-18-3, 98%) and ferulic acid (C_10_H_10_O_4_, M.W. 194.19, Cas. 1135-24-6, 98%) were obtained from Aladdin Chemistry Co. Ltd. (Shanghai, China). 3′-Hydroxy puerarin (C_21_H_20_O_10_, M.W. 432.38, Cas. 117060-54-5, 98%) was purchased from Shanghai PureOne BioTech. Co. Ltd. (Shanghai, China). Pratensein (C_16_H_12_O_6_, M.W. 300.26, Cas. 2284-31-3, 98%) was purchased from Wuhan ChemFaces Biotech Co., Ltd. (Wuhan, China). Methanol and water at mass spectrum purity grade were purchased from Merck KGaA (Darmstadt, Germany). All other reagents used in this study were purchased at analytical grade from the Guangzhou Chemical Reagent Factory (Guangzhou, China).

### 3.3. Preparation of Lyophilized Aqueous Extract from Naodesheng Tablet and Authentic Standard Solution

To avoid the possible solvent effect [51], *Naodesheng* Tablet was extracted using distilled water. The extract was lyophilized using a freeze dryer (FDU-1200, Eyela Co., Ltd., Shanghai, China) to prepare a lyophilized powder of *Naodesheng* Tablet (LNT). The whole process consulted the previous method [52,53] and is summarized in Figure 5.

The LNT sample was re-dissolved using methanol under ultrasound treatment and then filtered through a 0.45 μm membrane to prepare the sample solution (~30 mg/mL) [54,55]. Similarly, each authentic standard was also dissolved using methanol under ultrasound treatment and then filtered through a 0.45 μm membrane to obtain a standard solution (~10 μg/mL). The sample solution and all standard solutions were kept in a refrigerator (4 °C) for the following analyses. Similar to *Naodesheng* Tablet, 5 adulterated Tablets (i.e., CNT 1~CNT 5) were, respectively, treated by the above procedure as well. 

### 3.4. UHPLC-Q-Orbitrap MS Identification

#### 3.4.1. Chromatography and Mass Spectrometer Conditions

The UHPLC system (Thermo Fisher Scientific, Waltham, MA, USA) was equipped with an Accucore RP-MS LC C_18_ column (100 mm × 2.1 mm, 2.6 μm, Thermo Fisher, Waltham, MA, USA) for chromatographic separations. The mobile phase consisted of A (0.1% formic acid in water) and B (methanol) at a flow rate of 0.4 mL min^−1^ for the negative model. Under the positive model, phase A was replaced by 0.1% formic acid in water containing 5 mmol/L ammonium acetate and phase B was still methanol. The gradient elution was set as follows: 0–5 min, 10% B; 5–14.5 min, 10–100% B; 14.5–16 min, 100% B; 16–16.1 min, 100–10% B; 16.1 min–20 min, 10% B. The column temperature was maintained at 40 °C and the injection volume was 3 μL [56].

The above UHPLC system was coupled with a high-resolution Q-Orbitrap mass spectrometer (Thermo Fisher Scientific, Waltham, MA, USA). The operating parameters were set as follows: auxiliary gas, 10; sheath gas, 40; sweep gas, 0; spray voltage, 4.5 kV. The temperatures of the auxiliary gas heater and capillary were both set at 450 °C. The full MS resolution and data-dependent MS^2^ (dd-MS^2^) were 70,000 and 17500, respectively, while their automatic gain control (AGC) target was 2 × 10^5^. Nitrogen (N_2_) was applied for spray stabilization and the damping gas in the C-trap. The stepped normalized collision energy was set to 20, 50, and 90 V [57].

#### 3.4.2. Software, Data Acquisition, and Putative Identification

The Xcalibur 4.1 software package and TraceFinder General Quan (Thermo Fisher Scientific Inc., Waltham, MA, USA) were used for data acquisition and analysis. The acquired data included retention time, molecular peak, MS/MS profile, and typical fragments of authentic standards [58]. The data were recorded in the software package to build up a database of authentic standards. The data acquisition conditions were set as follows: 100–1500 Da mass range; 5 ppm mass tolerance; 5 S/N threshold; 10 min R.T. window override; 90% isotopic pattern fit threshold. The data of samples were acquired in the software package under the same conditions. Through the comparison, the bioactive compounds from the sample solution were preliminarily identified. After manual elucidation of MS spectrum fragmenting, the bioactive compounds were further confirmed to finish the putative identification.

#### 3.4.3. Semi-Quantification of Re-Nominated Q-Markers

The semi-quantification analyses of 5 re-nominated Q-markers (puerarin **12**, ginsenoside Rg1, HSYA **7**, citric acid **2**, and levistilide A **65**) were based on the principle of a previous study with minor modifications [59]. Briefly, the linear regression equation was first established through the injection of authentic standard solutions at different volumes into the UHPLC-Q-Exactive-Orbitrap MS system. The equipped Xcalibur 4.1 software offered peak area parameters for these authentic standard solutions. Under the same chromatography and MS spectrum conditions, sample solutions of certified and adulterated *Naodesheng* Tablets were subsequently injected into the system. According to the linear regression equation and peak area of the Q-markers, their chemical contents were finally quantified and expressed as mean ± SD. 

### 3.5. UV-Vis Spectrum Scanning

The UV-vis spectrum scanning of citric acid (**2**), HSYA (**7**), puerarin (**12**), notoginsenoside R1 (**36**), ginsenoside Rg1 (**39**), ginsenoside Rb1 (**50**), and levistilide A (**65**) was conducted based on a previous method [57]. In brief, citric acid (**2**) was dissolved in methanol to prepare the solution at 2 mg/mL. Others were dissolved in methanol to prepare the solution at 0.04~0.20 mg/mL, respectively. The solutions were individually analyzed by UV-vis spectrum scanning on a UV spectrophotometer (UV-2600A, UNICO, Co., Ltd., Shanghai, China) using methanol as a blank. The wavelength range and scanning accuracy were 195~1100 nm and 1 nm, respectively. The UV-vis spectrum scanning of each compound was performed three times in parallel.

### 3.6. Adulteration Detection Validation Experiment Based on Low-Version LC-MS Analysis

The quantum chemical calculations of 5 compounds, including notoginsenoside R1 (**36**), puerarin (**12**), HSYA **7**, citric acid (**2**), and levistilide A (**65**), were conducted with the B3LYP-D3 (BJ)/6-311G (d, p) basis set. The calculation tried to obtain the results of molecular geometry optimization, frequency calculation, and individual-point energy (SPE). The lack of an imaginary frequency was used to guarantee the optimal structure at the local minimum. The Gaussian 16 C.01 program was used to calculate the dipole moment and molecular polarity index (MPI) to characterize the molecular polarity degree [60,61,62,63].

### 3.7. Computational Details

The so-called “low-version LC-MS” referred to UHPLC-ESI-Q-TOF-MS analysis. It was used to validate whether the recommended Q-markers could detect the adulterated *Naodesheng* Tablets. Five Tablets were prepared through replacement by wood powder and named CNT 1~CNT 5, which characterized the defaults of *Gegen*, *Sanqi*, *Honghua*, *Shanzha*, and *Chuanxiong,* respectively (Table 5).

In brief, the Q-TOF-MS analysis was performed on a Triple TOF 5600*^plus^* mass spectrometer (AB SCIEX, Framingham, MA, USA) equipped with an ESI source, which was run in the negative ionization mode. The scan range was set at 100–2000 Da. The system was run with the following parameters: ion spray voltage, −4500 V; ion source heater temperature, 550 °C; curtain gas pressure (CUR, N_2_), 30 psi; nebulizing gas pressure (GS1, Air), 50 psi; Tis gas pressure (GS_2_, Air), 50 psi. The declustering potential (DP) was set at −100 V, whereas the collision energy (CE) was set at −45 V with a collision energy spread (CES) of 15 V. The above Q-TOF-MS system was connected with an ultra-high-performance liquid chromatography (UHPLC) system. The UHPLC system was equipped with a Phenomenex Luna C_18_ column (2.1 mm i.d. × 100 mm, 1.6 μm, Phenomenex Inc., Torrance, CA, USA). The mobile phase was employed for the elution of the system and consisted of a mixture of methanol (phase A) and 0.1% formic acid in water (phase B). The column was eluted at a flow rate of 0.2 mL/min with the following gradient elution program: 0–2 min, maintained at 30% B; 2–10 min, 30–0% B; 10–12 min, 0–30% B. The sample injection volume was set at 3 μL and the sample solution was 30 mg/mL.

The above experimental procedures were repeated using certified *Naodesheng* Tablet (Lot. 210803). Its sample injection volume was 3 μL and the sample solution was 30 mg/mL. The results of certified *Naodesheng* Tablet were compared with adulterated ones, to judge whether the Q-marker candidates could be used for adulteration detection.

### 3.8. Statistical Analysis

Each quantitative assessment experiment was performed in triplicate. The data were shown as the mean ± SD from three independent measurements. The calculation of correlation coefficients (R values) was based on linear analysis using Origin 6.0 professional software (Origin-Lab Corporation, Northampton, MA, USA).

## 4. Conclusions

In conclusion, by means of standards-library-dependent UHPLC-Q-Orbitrap MS putative identification, *Naodesheng* Tablet is evidenced to enrich 68 bioactive compounds. Of 68 identified compounds, HSYA, citric acid, levistilide A, puerarin, and notoginsenoside R1 are recommended to be included in the new Q-markers system. The LC-MS analysis of puerarin, notoginsenoside R1, HSYA, citric acid, and levistilide A can effectively detect adulterants regarding *Gegen*, *Sanqi*, *Honghua*, *Shanzha*, and *Chuanxiong* in *Naodesheng*.

## Figures and Tables

**Figure 1 molecules-29-01392-f001:**
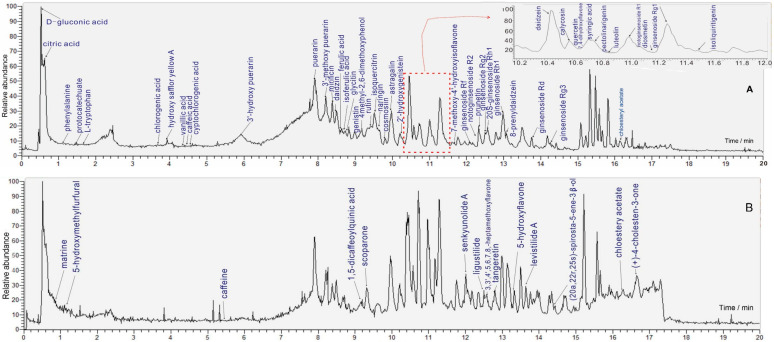
The TIC diagrams of *Naodesheng* Tablet in the UHPLC-Q-Orbitrap MS identification under negative mode (**A**) and positive mode (**B**). The positive mode was the supplement for the negative mode.

**Figure 2 molecules-29-01392-f002:**
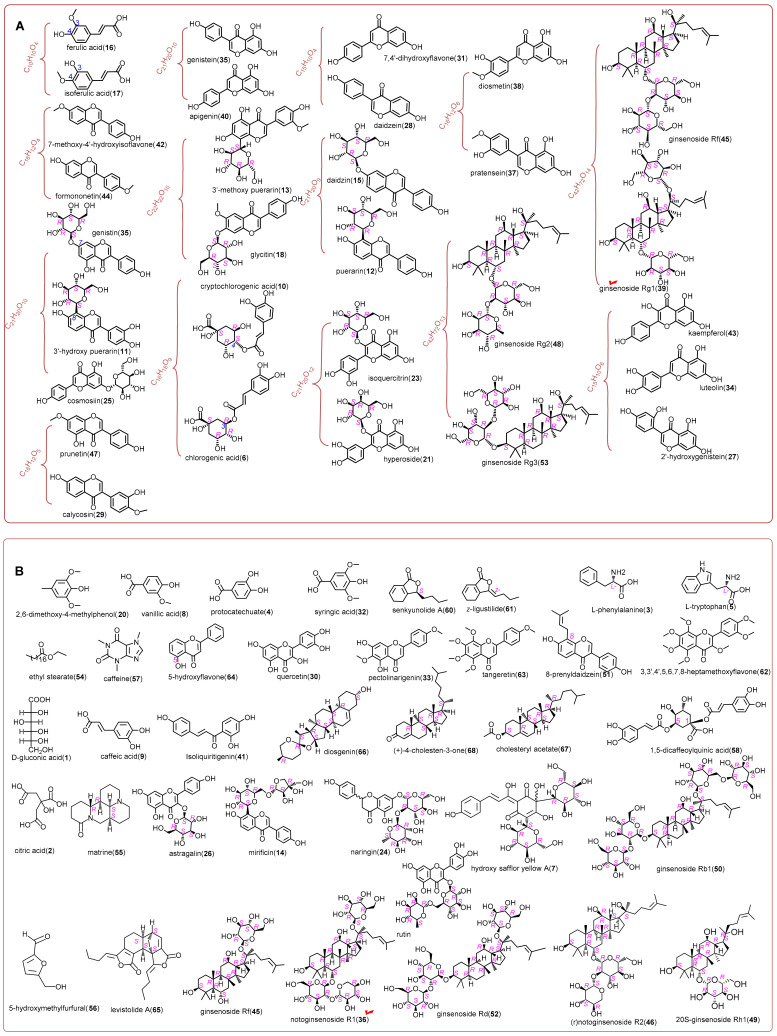
Structures of identified bioactive compounds from *Naodesheng* Tablet (**A**) for isomers; (**B**) for non-isomeric compounds. The chiral atoms in all sugar residue groups have been marked in their absolute configurations to avoid possible misreading. D-glucose is expressed as the Fischer project formula. The wave line in HSYA (**7**) indicates uncertain stereo configuration. The red tick √ means the old Q-markers.

**Figure 3 molecules-29-01392-f003:**
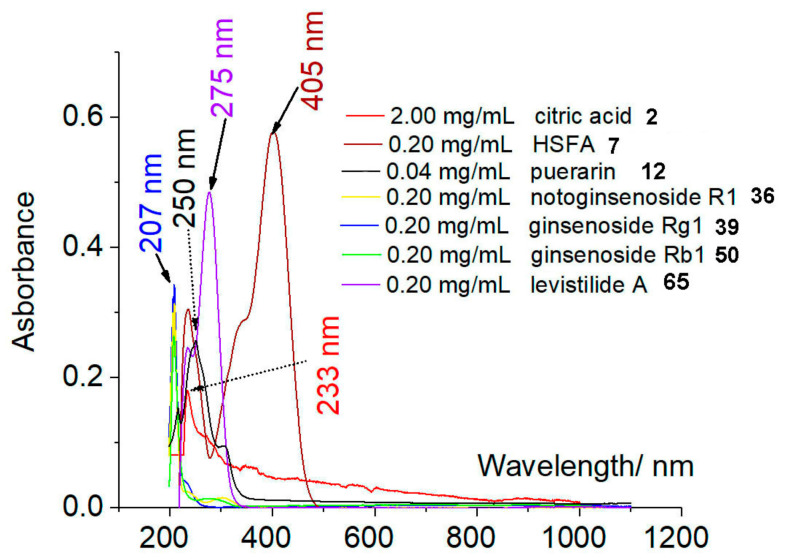
The UV-vis spectra of 7 compounds (**2**, **7**, **12**, **36**, **39**, **50**, and **65**).

**Figure 4 molecules-29-01392-f004:**
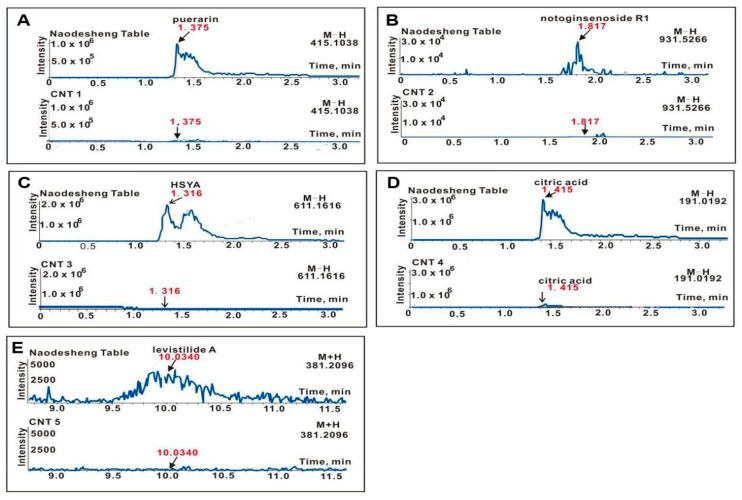
The results of the adulteration detection validation experiment of CNT 1~CNT 5. (**A**) *Naodesheng* Tablet and CNT 1 by extraction of C_21_H_19_O_6_ (puerarin [ M − H], *m*/*z* 415); (**B**) *Naodesheng* Tablet and CNT 2 by extraction of C_47_H_79_O_18_ (notoginsenoside R1 [ M − H], *m*/*z* 931); (**C**) *Naodesheng* Tablet and CNT 3 by extraction of C_27_H_31_O_16_ (HSYA [ M − H], *m*/*z* 611); (**D**) *Naodesheng* Tablet and CNT 4 by extraction of C_6_H_7_O_7_ (citric acid [ M − H], *m*/*z* 191); (**E**) *Naodesheng* Tablet and CNT 5 by extraction of C_24_H_29_O_4_ (levistilide A [ M + H], *m*/*z* 381). The analytic technology was UHPLC-ESI-Q-TOF-MS. (**A**–**D**) Under the negative model; (**E**) under the positive model.

**Figure 5 molecules-29-01392-f005:**
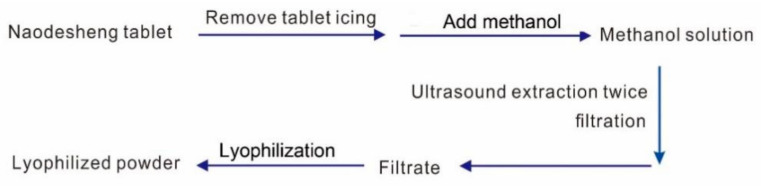
The preparation of the lyophilized aqueous extract of *Naodesheng* Tablet.

**Table 1 molecules-29-01392-t001:** The information of *Naodesheng* Tablet and 5 relevant herbal medicines.

Herbal Medicine	Plant Materials	Weight	Pharmacopoeia Q-Marker and Relevant Analytic Tool
*Naodesheng* Tablet (腦得生片)		665 g	ginsenoside Rg1 and Rb1, notoginsenoside R1 (HPLC); puerarin (TLC); ferulic acid (TLC)
*Gegen* (葛根)	Radix of *Pueraria lobata* (Willd.) Ohwi	261 g	puerarin (HPLC)
*Sanqi* (三七)	Radix or Rhizoma of *Panax notoginseng* (Burk.) F. H. Chen	78 g	ginsenoside Rg1 and Rb1, notoginsenoside R1 (HPLC)
*Honghua* (红花)	Dried flower of *Carthamus tinctorius* L.	91 g	HSYA (HPLC)
*Shanzha* (山楂)	Dried fruit of *Crataegus pinnatifida* Bunge	157 g	citric acid (HPLC)
*Chuanxiong* (川芎)	Rhizoma of *Ligusticum chuanxiong* Hort.	78 g	levistilide A (TLC) ferulic acid (HPLC)

Note: TLC, thinner-layer chromatography; HPLC, high-performance liquid chromatography.

**Table 3 molecules-29-01392-t003:** The main information of 68 putatively identified bioactive compounds (**1–68**) from *Naodesheng* Tablet).

	Reconstructed Q-Markers				
	Citric Acid (2) [1]	HSYA (7) [43,44]	Puerarin (12) [6,7,8]	NGR1 (36) [45]	levistilide A (65) [1]
Traceability	√	√	√	√	√
Specificity	√	√	√	√	√
Testability	√	√	√	√	√
Efficiency relevance	√	√	√	√	√
TCM relevance	√	√	√	√	√
Characterized herbal medicines	*Shanzha*	*Honghua*	*Gegen*	*Sanqi*	*Chuanxiong*

Note: HSYA, hydroxy safflor yellow A; NGR1, notoginsenoside R1.

**Table 4 molecules-29-01392-t004:** Semi-quantification results and computational chemistry results (including dipole moment value and HOMO→LUMO energy gap values of Q-marker candidates).

Q-Markers	Semi-Quantification/(%)	Computational Chemistry	
		Dipole Moment	HOMO → LUMO
citric acid (**2**)	0.822 ± 0.021	2.0819	687.3211
HSYA (**7**)	0.039 ± 0.002	6.6315	308.3945
puerarin (**12**)	1.044 ± 0.176	1.9418	405.7099
notoginsenoside R1 (**36**)	0.128 ± 0.001	7.2955	680.0210
levistilide A (**65**)	0.070 ± 0.006	5.7291	402.9922

Note: The semi-quantification was based on the certified and adulterated *Naodesheng* Tablet using UHPLC-Q-Orbitrap MS/MS analysis and its results were expressed as mean ± standard deviation (SD) (*n* = 3). The relevant data are detailed in Appendix A. The computational chemistry was conducted using a restricted B3LYP basis set. Dipole moment value, Debye unit; HOMO → LUMO, the energy gap from the highest occupied molecular orbital to the lowest unoccupied molecular orbital, kJ/mol unit.

**Table 5 molecules-29-01392-t005:** Five Q-marker candidates for detecting the corresponding adulterated *Naodesheng* Tablets (CNT1~CNT5).

Name	*Gegen*	*Sanqi*	*Honghua*	*Shanzha*	*Chuanxiong*	Q-Marker for Analysis
CNT 1	wood	√	√	√	√	puerarin (**12**)
CNT 2	√	wood	√	√	√	notoginsenoside R1 (**36**)
CNT 3	√	√	wood	√	√	HSYA (**7**)
CNT 4	√	√	√	wood	√	citric acid (**2**)
CNT 5	√	√	√	√	wood	levistilide A (**65**)
Certified Tablet	√	√	√	√	√	HSYA (7), puerarin (**12**), notoginsenoside R1 (**36**), levistilide A (**65**)

## Data Availability

Dataset available on request from the authors.

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
