# Peer review of "Detection of Adulterated Naodesheng Tablet (Naodesheng Pian) via In-Depth Chemical Analysis and Subsequent Reconstruction of Its Pharmacopoeia Q-Markers"

_molecules, 2024, doi:10.3390/molecules29061392_

Round 1

Reviewer 1 Report

Comments and Suggestions for Authors

Several errors: use inapppropriate of italic (lines 45, 77 etc..); missed bracket (line 172); missed information (lines 169-170, 193, 207); mathematic notation (line 203)

Check and correct references style in the manuscript. 

Justify the text in all manuscript

Better resolution of Figure 2,4,5,6,7,8

Check the order of Figures in the manuscript (Figure 8 before Figures 6 and 7, two Figures indicated as 8, no caption for Figure at pag 16 )

Comments on the Quality of English Language

Quality of Englisl can be improved

Author Response

Comments and Suggestions for Authors

Several errors: use inapppropriate of italic (lines 45, 77 etc..).

→ Corrected. Please kindly see: Line 16-17, 45, 54, 78. 

missed bracket (line 172).

→ Corrected. Please kindly see: Line 348. 

missed information (lines 169-170, 193, 207).

→ Corrected. Please kindly see: Line 346-347, 371, 378, 388. 

mathematic notation (line 203)

→ Corrected. Please kindly see: Line 382. 

Check and correct references style in the manuscript.

→ Carefully checked and revised. Please kindly see: Line 487-624. 

Justify the text in all manuscript. Better resolution of Figure 2,4,5,6,7,8

→ Figure 2: It has high resolution with 1200 value.

Figure 3: re-drawn and removed into Suppls. 27, 34, 43.

Figure 5: re-drawn and removed into Suppls. 39, 45.

Figure 6: re-drawn and removed into Suppls. 48, 53.

Figures 7: re-drawn and removed into Suppls. 29, 47.

Check the order of Figures in the manuscript (Figure 8 before Figures 6 and 7, two Figures indicated as 8, no caption for Figure at page 16 )

→ Checked and corrected.

Comments on the Quality of English Language

Quality of Englisl can be improved

→ “Englisl” should be “English”

→ Improved.

Reviewer 2 Report

Comments and Suggestions for Authors

I have reviewed your paper titled "Detection of adulterated Naodesheng Tablets via in-depth chemical analysis and subsequent reconstruction of its pharmacopoeia quality-markers". The paper reports the identification of 68 compounds from herbal medicines in Naodesheng Tablets, and the re-nomination of five quality-markers for the quality control and adulteration detection of the formula. The paper uses a novel strategy of standards-library-dependent UHPLC-Q-Orbitrap MS/MS putative identification, and provides valuable recommendations for the Pharmacopoeia Commission. However, I have some major concerns and suggestions that need to be addressed before the paper can be accepted for publication.

- The authors should reduce the data (Tables and Figures) presented in the paper. The paper contains too many tables and figures, which makes it difficult to follow and comprehend. Some of the tables and figures are redundant. The authors should consider merging, deleting, or moving some of the tables and figures to the supplementary material, and only keep the essential ones in the main text.

- The compound numbers or IDs should be added in Table 3 and Figure 3. Table 3 and Figure 3 show the MS chemical calculation results of the isomers, respectively. However, the compound numbers or IDs are not given in these tables and figures, which makes it hard to correlate them. The authors should add the compound numbers or IDs in Table 3 and Figure 3.

- Table 4 should provide the quantitative data of the re-nominated quality-markers. Table 4 shows the qualitative data of the re-nominated quality-markers in the certified and adulterated Naodesheng Tablets, such as the retention time, the mass accuracy, and the MS/MS fragments. However, the quantitative data, such as the peak area, the concentration, and the relative content, are not given in this table. The authors should provide the quantitative data of the re-nominated quality-markers in Table 4, and explain how they were calculated and normalized.

I hope that the authors will consider these comments and suggestions and revise their paper accordingly. I look forward to seeing the revised version of the paper.

Author Response

Comments and Suggestions for Authors

I have reviewed your paper titled "Detection of adulterated Naodesheng Tablets via in-depth chemical analysis and subsequent reconstruction of its pharmacopoeia quality-markers". The paper reports the identification of 68 compounds from herbal medicines in Naodesheng Tablets, and the re-nomination of five quality-markers for the quality control and adulteration detection of the formula. The paper uses a novel strategy of standards-library-dependent UHPLC-Q-Orbitrap MS/MS putative identification, and provides valuable recommendations for the Pharmacopoeia Commission. However, I have some major concerns and suggestions that need to be addressed before the paper can be accepted for publication.

  • The authors should reduce the data (Tables and Figures) presented in the paper. The paper contains too many tables and figures, which makes it difficult to follow and comprehend. Some of the tables and figures are redundant. The authors should consider merging, deleting, or moving some of the tables and figures to the supplementary material, and only keep the essential ones in the main text.

→ Figs. 3-6 are removed into supplementary material, i.e., Suppls. 27, 29, 34, 39, 43, 45, 47, 48, and 53.

  • The compound numbers or IDs should be added in Table 3 and Figure 3. Table 3 and Figure 3 show the MS chemical calculation results of the isomers, respectively. However, the compound numbers or IDs are not given in these tables and figures, which makes it hard to correlate them. The authors should add the compound numbers or IDs in Table 3 and Figure 3.

→ The IDs have been added. Please kindly see: Table 2, Figure 2, Table 3, Figure 3, Table 4, and main text.

  • Table 4 should provide the quantitative data of the re-nominated quality-markers. Table 4 shows the qualitative data of the re-nominated quality-markers in the certified and adulterated Naodesheng Tablets, such as the retention time, the mass accuracy, and the MS/MS fragments. However, the quantitative data, such as the peak area, the concentration, and the relative content, are not given in this table. The authors should provide the quantitative data of the re-nominated quality-markers in Table 4, and explain how they were calculated and normalized.

→ This of course is a reasonable comment.

   Quantification of the re-nominated Q-markers: added in Table 4. Relevant experiments were added in 3.4.3 Section and 3.8 Section. The quantification however is calculated using external standard method rather than normalization method. The linear regression equation, R value and linear range of authentic standard as well as the peak area (mean ± SD) of tested sample are detailed in Suppl. 69. In addition, a calibration curve diagram is also followed. The authentic standards refer to puerarin (12), notoginsenoside R1 (36), HSYA (7), levistilide A (65), and citric acid (2). The tested sample is certified and adulterated Naodesheng Tablet. However, it should be emphasized that, the current LC-MS technology, UHPLC-Q-Orbitrap MS/MS, is highly accurate in qualitative analysis; however, it is not so accurate quantitative analysis. This is its sign response is based on the mass itself, thus, in low concentration, its peak signs (e.g., peak height and peak area) usually have substantial uncertainty. In R values of regression equation varied from 0.9906 to 0.9994, and thus we described as “Semi-quantification”. The quantification result is correspondingly added into Table 4. Please kindly see: Line 29, New Table 4, and Suppl. 69.

   The retention time (R.T.): This is included in the previous version in Table 2 now.

The mass accuracy. (1) This has been already mentioned in the previous version (see: new Line 119-124). (2) Now, we further add the “Error δ (ppm)” value in new Table 2. Meanwhile, we also mentioned them in new Line 123-126.

The MS/MS fragments: This has been already listed in the previous version (see: Table 2).

  • I hope that the authors will consider these comments and suggestions and revise their paper accordingly. I look forward to seeing the revised version of the paper.

→ Thank you very much!

Round 2

Reviewer 2 Report

Comments and Suggestions for Authors

Upon reviewing the revised manuscript titled "Detection of Adulterated Naodesheng Tablet via In-depth Chemical Analysis and Subsequent Reconstruction of its Pharmacopoeia Q-markers," I would like to acknowledge the comprehensive revisions made in accordance with the previous comments.

The manuscript has addressed all the concerns raised during the initial review process effectively.

The clarity and depth of the research, along with the validation experiments, solidify the manuscript's contribution to ensuring the authenticity of traditional Chinese medicine.

I recommend the paper for acceptance and believe it will offer valuable insights to the Pharmacopoeia Commission and the broader scientific community.